# Antibiotic prescription practices among prescribers for children under five at public health centers III and IV in Mbarara district

Nelson Okello [1,2]*, Joseph Oloro [3], Catherine Kyakwera[1], Elias Kumbakumba[1], Celestino Obua[3]

1 Faculty of Medicine, Department of Paediatrics and Child Health, Mbarara University of Science and Technology, Mbarara, Uganda, 2 Faculty of Health Sciences, Department of Paediatrics and Child Health, Lira University, Lira, Uganda, 3 Faculty of Medicine, Department of Pharmacology & Therapeutics, Mbarara University of Science and Technology, Mbarara, Uganda

* docnelsons@gmail.com

## Abstract

### Introduction

Rational use of medicines requires that patients receive medications appropriate to their clinical needs. Irrational prescription of antibiotics has been reported in many health systems across the world. In Uganda, mainly nurses and assistant medical officers (Clinical officers) prescribe for children at level III and IV primary care facilities (health center II and IV). Nurses are not primarily trained prescribers; their antibiotic prescription maybe associated with errors. There is a need to understand the practices of antibiotic prescription among prescribers in the public primary care facilities. We therefore determined antibiotic prescription practices of prescribers for children under five years at health center III and IV in Mbarara district, South Western Uganda.

### Methods

This was a retrospective descriptive cross-sectional study. We reviewed outpatient records of children <5 years of age retrospectively. Information obtained from the outpatient registers were captured in predesigned data abstraction form. Health care providers working at health centers III and IV were interviewed using a structured questionnaire. They provided information on socio-demographic, health facility, antibiotic prescription practices and availability of reference tools. Data was analyzed using STATA software version 13·0.

### Results

There were 1218 outpatients records of children under five years reviewed and 35 health care providers interviewed. The most common childhood illness diagnosed was upper respiratory tract infection. It received the most antibiotic prescription (53%). The most commonly prescribed oral antibiotics were cotrimoxazole and amoxicillin, and ceftriaxone and benzyl penicillin were the commonest prescribed injectable antibiotics. Up to 68.4% of the antibiotic prescription was irrational. No prescriber or facility factors were associated with irrational antibiotic prescription practices.

**Data Availability Statement:** All relevant data are within the manuscript and its Supporting Information files.

**Funding:** This study was funded by Makerere University – Swedish International development Agency (SIDA) research scholarship. The funders had no role in study design, data collection and analysis, decision to publish, or preparation of manuscript.

**Competing interests:** NO. The authors have declared no competing interest exist.

**Abbreviations:** EDL, Essential Drug List; HC, Health Center; IMNCI, Integrated Management of Newborn and Childhood Illnesses; LHD, Local Health District; MAK, Makerere University; MOH, Ministry of Health; MUST, Mbarara University of Science and Technology; SIDA, Swedish International Development Cooperation Agency; UCG, Uganda Clinical Guideline; URTI, Upper Respiratory Tract Infection; VHT, Village Health Team.

## Conclusion

Upper respiratory tract infection is the most diagnosed condition in children under five years with Cotrimoxazole and Amoxicillin being the most commonly prescribed antibiotics. Antibiotics are being prescribed irrationally at health centers III and IV in Mbarara District. Training and support supervision of prescribers at health centers III and IV in Mbarara district need to be prioritized by the district health team.

## Introduction

Medicines play an integral part of healthcare delivery [1]. However, they are expensive commodities and account for a significant proportion of overall health expenditure in most countries. Rational use of medicines requires that patients receive medications appropriate to their clinical needs, in doses that meet their own individual requirements, for an adequate period of time and at the lowest cost to them and their community [2]. Irrational use of medicines described as the medically inappropriate and economically ineffective use of pharmaceuticals is a major challenge facing many health systems across the world [1].

Children represent a subset of the population who frequently receive antibiotics. Antibiotics are the most commonly prescribed drugs. It is particularly necessary after a confirmation culture and sensitivity studies, but structurally primary care units do not have facilities for microbial studies. Approximately three quarters of all outpatient antibiotic prescriptions given to children are for treatment of upper respiratory tract conditions [3]. Most children are seen in a primary health care setting by primary care givers, mainly nurses, assistant medical officers (clinical officers) and occasionally medical officers. Clinical officers are health cadres with diploma in medicine and community health, they work under supervision of holders of bachelor of medicine and bachelor of surgery (Medical officers/General Practioners).

Uganda's healthcare system works on a referral basis. Village health teams (VHT) work in the community at the lowest health care level where they provide referral of patients from the community; the second level of care often called health center II (dispensaries) are located at a parish level, health center IIIs and IVs are located at a sub county and a county level respectively. Health center III provide mainly maternal health and ambulatory care services. Health center IV has an operation room and provide in-patient and ambulatory services. Patients from a health center IV are referred to the district hospital, located at the district level. Regional and National Referral hospital are located at the Regional and National level respectively. Health centers III and IV (level III and IV) lack capacity to perform most of the investigations including culture/sensitivity testing and X-rays.

To address this and improve prescription practices, Uganda has a national clinical guideline (UCG) [4] last updated in 2016. It aimed at providing easy-to-use, practical, complete, and useful information on how to correctly diagnose and manage all common conditions in a primary care setting. The clinical guide should help in making the most appropriate use of scarce diagnostic and clinical resources, including medicines. The Regional Referral hospital and the district hospitals according to the Uganda health systems should be able to provide supervision role and mentorship to the lower health facilities within the district.

Health workers at health centers III and IV depend on their clinical skills and experience in making decision about antibiotic prescriptions. Nurses are not primarily trained as prescribers and thus their antibiotic prescription may be associated with errors. There is scarce information about the practice of antibiotic prescription among prescribers in public health centers III

and IV in Mbarara District, South Western Uganda. We therefore evaluated pediatrics antibiotic prescription practices among the health workers caring for children under five years at the primary health care settings in Mbarara district.

## Materials and methods

### Study design

This was a descriptive cross-sectional study of antibiotic prescription practices among prescribers of children under five years at 17 public Health center III and IV in Mbarara District. This was conducted from February to May 2019.

### Study setting

Mbarara district is found in the South-Western region of Uganda. It has a Regional Referral hospital, 13 Health Center III and four Health Centers IV. Most of these facilities are located in the rural and peri-urban areas and receive medical supplies from Uganda national medical store based on Uganda national essential drug list. The antibiotics supplied to these facilities is based on the level of care and must be on the essential drug list. They include among others, amoxicillin, erythromycin, azithromycin, cotrimoxazole, ampicillin, cloxacillin, ceftriaxone, benzyl penicillin and ciprofloxacin.

The health center III has about 19 health staff, headed by a Senior Clinical Officer (Senior assistant medical officer), it has an outpatient, laboratory and maternity services and serves about 23,969 people of which about 5033 (21%) are children under five [5]. Health centers IV serves a county or a parliamentary constituency, it is managed by two medical officers and other cadres; it serves a population of about 100,000 of which 21,000 (21%) are children under 5years of age. Health center III and IV facilities have limited investigation capacities, for instance, they do not have an X-ray machine and they have no facility for doing culture and sensitivity testing. Most often, prescribers at public health centers III and IV use the Uganda clinical guideline and manual for integrated management of newborn and childhood illnesses [6] for diagnosis and treatment of common childhood illnesses.

### Study population

The study included 35 health care workers attending to children at Health center III and IV facilities and outpatient registers of children under five years treated with antibiotics within the last six months in the facilities.

### Eligibility criteria

Prescribers (nurses, midwives, clinical officers and medical officers) at public health centers III and IV.

Records with complete entries.

### Sample size and sampling procedure

The Cochran's formula for sample size determination of a single population proportion [7], was used to calculate sample size for prescribers in this study. The prevalence of irrational antibiotic prescriptions among health care workers in lower health facilities in Mbarara was assumed at 50%. This was an arbitrary value given that there was no comparative data. 80% power and 95% confidence intervals were considered, and a sample size of 384 was realized for an infinite population, which was corrected for a finite population to 35. Participants were recruited consecutively until the sample size was reached. Each health center III and IV was

visited once during data collection and two prescribers from each facility were interviewed using a structured questionnaire. In order to determine the common childhood illnesses treated with antibiotics in Health center III and IV, we systematically reviewed the facilities outpatients' registers. The information obtained was entered into a data abstraction form. We classified the identified illnesses according to International Classification of Diseases 10 (ICD 10) [8] as indicated in the Uganda Clinical Guideline 2016. Conditions that matched integrated management of newborn and childhood illnesses protocol were classified based on the integrated management of Newborn and childhood illnesses manual [9].

In order to identify the most commonly prescribed antibiotics for common childhood illnesses in HC III and IV, we counted physically from the outpatient registers the number of children under five years of age who received antibiotic prescription for common childhood illnesses in the last 6 months using a predesigned data abstraction form. This included illnesses such as pneumonia, cough or cold, malaria, otitis media, urinary tract infection, diarrhea and dysentery as indicated on integrated management of newborn and childhood illnesses [9] manual. To decide whether the prescription was rational or not, we compared the diagnosis/classification with the standard guideline.

In order to establish health system and prescriber factors that affect antibiotic prescription practices, we conducted a face-to-face interview with each health unit in charge and health care providers who make prescription for children using a structured questionnaire in English. The interview covered areas of prescribers' characteristics (level of education, last qualification attained, work experience and experience in pediatric care), availability and utilization of Uganda clinical guideline, availability of antibiotics and common childhood illnesses that require antibiotic prescription and utilization of laboratory services to test children for malaria before antibiotic prescription.

## Quality control

A structured questionnaire guided by the Uganda clinical guideline 2016 was used in data collection. The Uganda clinical guideline 2016 is a validated tool used for primary care prescription [4]. Research assistants were trained on the data collection procedures, tools and ethical considerations of the study. Regular supervision was done to ensure consistency in the data being collected. Data check was done at the end of every day of data collection and stored for entry.

## Ethics and consent

Study protocols and procedures were reviewed and approved by Mbarara University of Science and Technology (Ref. 15/11-18). The study received clearance from the Uganda National Council for Science and Technology (Ref. SS4924). We sought a written permission from the office of the District Health Officer (DHO), Municipal Health Officer Mbarara district local government to enable research in the district and municipal health facilities. We obtained written informed consent from the health care providers and the facility in-charge; this was done in the local language (Runyankole) for good understanding. Participants were assured of privacy and confidentiality of the information provided by them.

## Data analysis

The data set was entered into Epi-Info software version 7·2 database for both data set. These were imported into STATA software version 13·0 for analysis.

We summarized baseline characteristics using frequencies and proportions for common childhood illnesses for which antibiotics was prescribed; common prescribed antibiotics. We

also generated variables; rational and irrational prescriptions and compared with prescription guidelines. A univariate analysis, Chi-square test and fisher's exact test was used to establish the relationships between the health system and prescriber factors with irrational antibiotic prescriptions for common childhood illnesses at significance level of 5%.

All factors with p-value <0.01 and those with a known biologically credible association with irrational antibiotic prescriptions were considered in the multivariate analysis which was performed to control confounding.

## Results

Of the 1218 records, 65.6% were from HC III and most of which were for children aged 6 months and above. Only 8.1% of the records were for children below 6 months old. Records of females were more than males (53.1%5 vs 46.9%) (Table 1). Of the 35 prescribers interviewed, about sixty nine percent (68.6%) were from health center III with more females (74.3%). Most of the prescribers (77.1%) were either nurses or midwives, majority of whom (61.76%) had either diploma or degree. Majority (57.1%) of the prescribers interviewed were aged below 35 years and 42.9% of them were aged 35 years and above. The mean age was 36.5 years. Only 22.9% of the prescribers were clinicians (Clinical officers and Medical officers). Most of the prescribers (77.1%) had experience in childcare for five or more years and up to 45.7% of the prescribers had been in practice for 10years and above. Almost no prescribers had received any specific training on management of common childhood illnesses in the last 6 months (Table 2).

Mostly Upper respiratory tract infection (URTI), Pneumonia and acute watery diarrhea received antibiotic prescription. Pneumonia was more likely to be diagnosed in HC IV (p < 0.05) while upper respiratory tract infection was more likely to be diagnosed in HC III (p = 0.01) (Fig 1).

Cotrimoxazole and Amoxicillin were the most prescribed oral antibiotics while Benzyl penicillin and ceftriaxone were the most prescribed injectable antibiotic. Two (2) children received doxycycline, an antibiotic contraindicated in this age group (Fig 2).

Almost all (97.1%) of the facilities had a reference tool, mainly Uganda Clinical Guideline and 62.9% had a Manual for Integrated Management of Neonatal and Childhood Illnesses (IMNCI 2012). Despite the presence of reference tools, 68.4% of the prescribed antibiotics was irrational as compared with the existing guideline. Irrational antibiotic prescriptions were more common at health centers III (73%) than in Health centers IV (59.8%) (P-value <0.05).

**Table 1. Demographic characteristics of child from records, N = 1218.**

| Characteristics | Frequency | Percentage |
|---|---|---|
| **Facility level** | | |
| **HC III** | 799 | 65.6 |
| **HC IV** | 419 | 34.4 |
| **Age in months, Mean (SD)** | | 14.2 |
| **Age categories in months** | | |
| **0–5** | 98 | 8.1 |
| **6–11** | 237 | 19.5 |
| **12–23** | 360 | 29.6 |
| **24–59** | 523 | 42.9 |
| **Gender** | | |
| **Male** | 571 | 46.9 |
| **Female** | 647 | 53.1 |

**Table 2. Table showing findings from prescribers' interview, N = 35.**

| Characteristics | Frequency | (%) |
|---|---|---|
| **Facility level** | | |
| HC III | 24 | 68.6 |
| HC IV | 11 | 31.4 |
| **Gender** | | |
| Male | 9 | 25.7 |
| Female | 26 | 74.3 |
| **Duration in paediatric care(years), mean (SD)** | | 10.1 (7.3) |
| **Duration in paediatrics care(years)** | | |
| <5 | 8 | 22.9 |
| 5–9 | 11 | 31.4 |
| > = 10 | 16 | 45.7 |
| **Designation** | | |
| **Medical officer** | 2 | 5.7 |
| **Clinical officer** | 6 | 17.1 |
| **Nurse/midwife** | 27 | 77.1 |
| **Highest level of training attained** | | |
| **Certificate** | 14 | 40 |
| **Diploma** | 18 | 51.4 |
| **Degree** | 3 | 8.6 |
| **Training received for management of specific childhood illness** | | |
| **Upper Respiratory Tract Infection** | 7 | 20.0 |
| **Pneumonia** | 8 | 22.9 |
| **Acute watery diarrhoea** | 8 | 22.9 |
| **Urinary Tract Infection** | 8 | 22.9 |
| **Otitis media** | 4 | 11.4 |
| **Bloody diarrhoea** | 6 | 17.1 |

Upper respiratory tract infection and acute watery diarrhea were the conditions that received most irrational antibiotic prescription (Fig 3).

Of the 35 prescribers interviewed, sixteen (16) reported having prescribed antibiotics for common childhood illnesses. Ten out of sixteen reported irrational antibiotic prescription, although there was no difference in prescription practices across prescribers/facility factors (Table 3).

## Discussion

This study was conducted to determine the antibiotic prescription practices of prescribers for children under five years at public health centers III and IV in Mbarara District. Results showed that Upper respiratory tract infection contributed more than half of the common childhood illness in Health centers III and IV. It also received the most antibiotic prescription. It is thought that Upper respiratory tract infection is mainly viral and antibiotic has no role in its management [10], a few cases could be due to a bacterial etiology. Our finding could be because diagnosis at this level of care is mainly clinical but also due to the fact that majority of the prescribers in our study were either nurses or midwives. A health cadre not primarily trained to make diagnosis and prescription. This might have led to a misclassification of illnesses. This is similar to findings of the studies in Ghana [11] and India where acute tonsillitis and otitis media (AOM) were as common as upper respiratory tract infection and represented

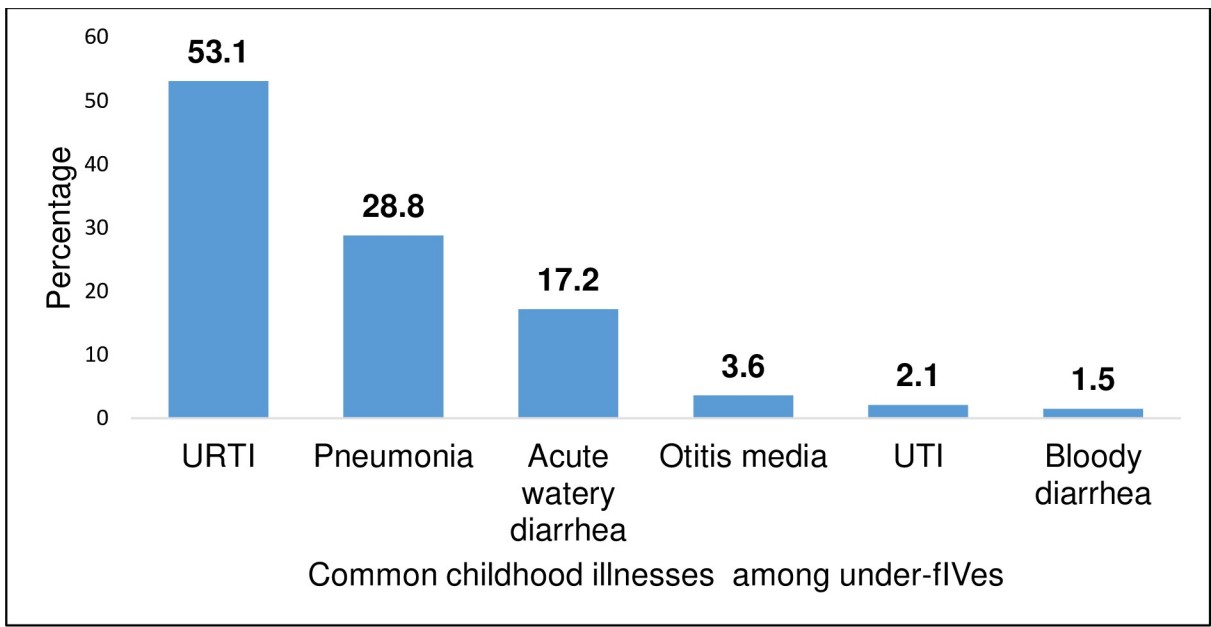

**Fig 1. Common childhood illnesses treated with antibiotics.**

more than 69% of all indications for prescribing antibiotics [12]. Our finding was however contrary to findings in United States, where two thirds of the viral upper respiratory tract

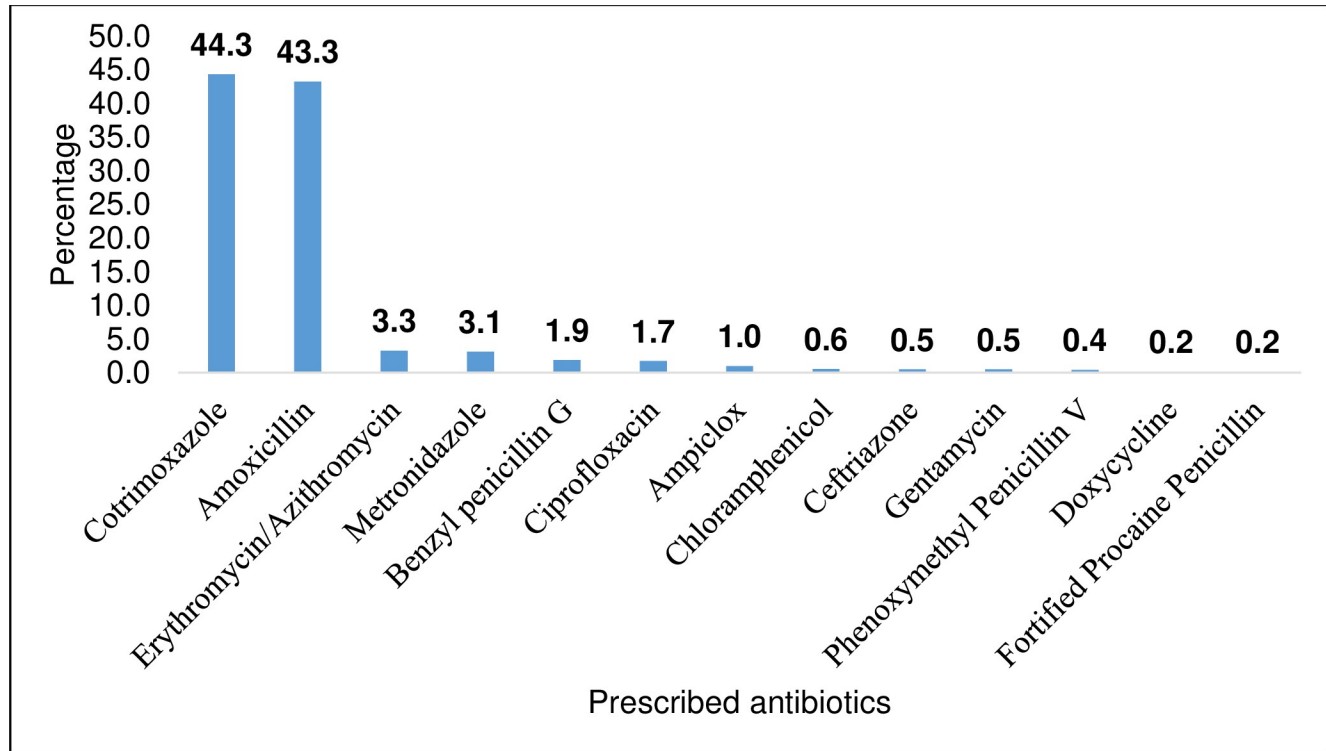

**Fig 2. Common antibiotic prescribed.**

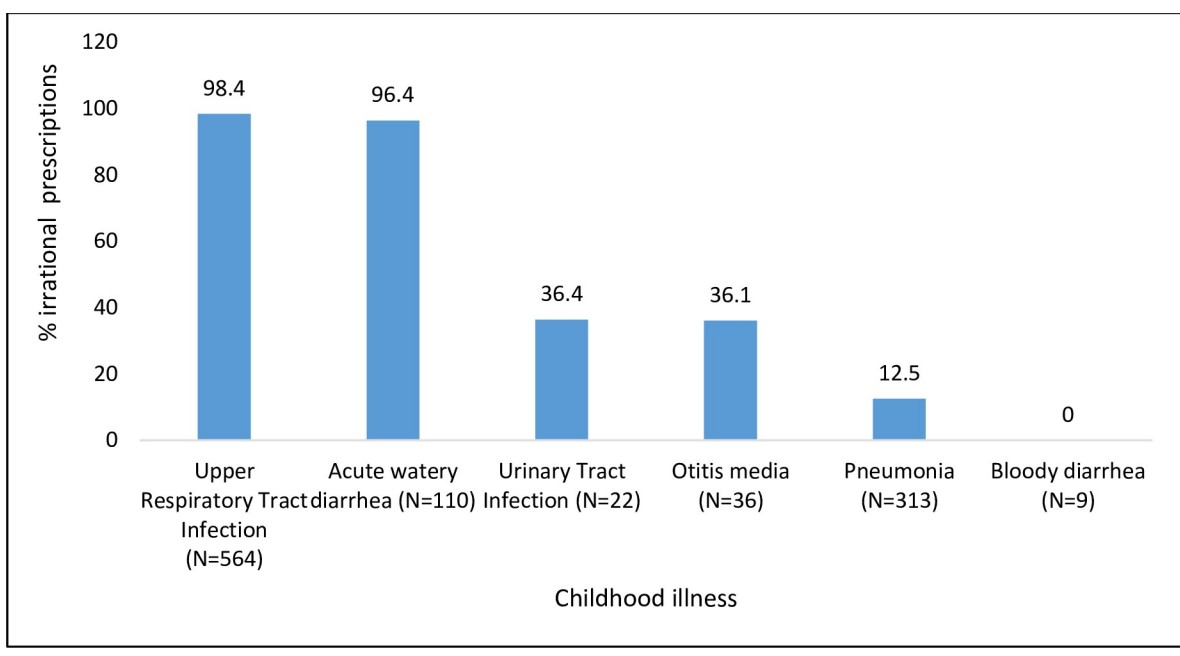

**Fig 3. Proportion of irrational antibiotic prescription based on the existing guideline.**

infection was treated with antibiotics [13]. A lower prevalence of 38% was observed in Netherlands [14].

Cotrimoxazole and Amoxicillin were prescribed in the majority of childhood illnesses such as acute watery diarrhoea in which antibiotic is not recommended [15] (Uganda Clinical Guideline, pages 335–336 & 383–385) [4]. This could be because Amoxicillin is a common medicine and it is generally cheap, therefore easily prescribed. Amoxicillin and Cotrimoxazole were more available in these facilities and prescribers were more likely to make prescription of what they had in stock. The over prescription of amoxacillin and cotrimoxazole in our study is not surprising. These medicines are recommended for management of uncomplicated bacterial infection in children. They are preferred first line and are readily available. It was however surprising that these antibiotics were also prescribed for acute watery diarrhea, a condition that is mainly viral. Similarly, in Ghana, Prah (2017) found amoxicillin prescription was at 22.5% [11]. While Kibuule (2016) in Uganda found amoxicillin over use of about 30 percent. Amoxacillin prescription in children is lower in some parts of the world. In Albania, Mihani and Kelici (2018) found a lower proportion of amoxicillin prescription of only 19.4% [16]. This could be due to the fact that in their setting, diagnostic confirmation is done before antibiotic prescription unlike in the Ugandan case where most prescription for out patients are made on an empirical basis. Although injectable antibiotic use in outpatients was also noted in our study, their prescription was generally low compared to that of oral antibiotics. This means that the prescribers in these facilities makes prescription from the essential drug list for Uganda, a practice that should be encouraged. Doxycicline was prescribed for children below 5 years of age. This drug is generally contra indicated in this age group due to its side effect of enamel hypoplasia and tooth pigmentation [17].This could be due to knowledge gap among the prescribers, considering that majority of the prescribers in our study were either nurses or midwives. These health care cadres are not primarily trained to prescribe according to the Ugandan guidelines.

About two-thirds of the prescribed antibiotic was irrational, yet all the facilities had the guideline that were accessible. Upper respiratory tract infection and or acute watery diarrhea

**Table 3. Relationship between irrational prescription and prescribers'/health facility factors, N = 16.**

| Prescribers Characteristics | Irrational Prescription Freq (%) | Rational Prescription Freq (%) | Fisher's Exact P-Value |
|---|---|---|---|
| **Age (yrs.)** | | | 0.738 |
| <35 (n = 10) | 5 (50.0) | 5 (50.0) | |
| 35–44 (n = 1) | 1 (100) | 0 (0.0) | |
| > = 45 (n = 5) | 4 (80.0) | 1 (20.0) | |
| **Gender** | | | 0.588 |
| Male (n = 5) | 4 (80.0) | 1 (20.0) | |
| Female (n = 11) | 6 (54.5) | 5 (45.5) | |
| **Work experience** | | | 0.091 |
| <5 (n = 4) | 3 (75.0) | 1 (25.0) | |
| 5–9 (n = 5) | 1 (20.0) | 4 (80.0) | |
| > = 10 (n = 7) | 6 (85.7) | 1 (14.3) | |
| **Education level** | | | 0.604 |
| Diploma or more (n = 12) | 8 (66.7) | 4 (33.3) | |
| Certificate (n = 4) | 2 (50.0) | 2 (50.0) | |
| **Training** | | | 0.299 |
| None (n = 11) | 8 (72.7) | 3 (27.3) | |
| At least one (n = 5) | 2 (40.0) | 3 (60.0) | |
| **Facility level** | | | 1.00 |
| III (n = 12) | 7 (58.3) | 5 (41.7) | |
| IV (n = 4) | 3 (75.0) | 1 (25.0) | |
| **Reference guide** | | | 0.375 |
| Not available (n = 1) | 0 (0.0) | 1 (100) | |
| Available (n = 15) | 10 (66.7) | 5 (33.3) | |
| **Accessibility of the guide** | | | 1.00 |
| Accessible (n = 15) | 1 (6.7) | 14 (93.3) | |
| Not Accessible (n = 1) | 1 (100) | 0 (0.0) | |
| **Drug stock out** | | | 0.588 |
| Stock out (n = 11) | 6 (54.5) | 5 (45.5) | |
| No stock out (n = 5) | 4 (80.0) | 1 (20.0) | |

were irrationally treated with antibiotics. These childhood illnesses are caused mainly by viruses and can be managed supportively [9]. This finding indicates lack of knowledge on the classification and management of common childhood illnesses, it could also reflect poor adherence to the guideline. This could be explained by the fact that majority of the prescribers in our study were not primarily trained as prescribers. It appears that the prescribers in our study made their prescriptions based on their previous experience or by copying others. Guidelines were not utilized in making antibiotic prescription. This finding was comparable to a Ghanaian and Chinese study by Prah (2017) and Song (2018) where more than half of antibiotic prescription was irrational. In Netherlands and Korea, irrational antibiotic prescription was generally low [18].

We did not find any relationship between prescription practices and prescribers/facility factors. This was surprising given the fact that most of our prescribers were either nurses or midwives who were not trained prescribers. This could be explained by the fact our study was not powered to assess the relationship or associations. The irrational antibiotic prescription was noted across the prescribers' and facility factors. This was contrary to a study done by A.

Kotwani (2012) on factors influencing primary care physicians to prescribe antibiotics in Delhi-India [3]. They found diagnostic uncertainty, lack of time due to overcrowding, consideration about suitability, laxity in regulation for prescribing and dispensing, self-medication and doctors' shopping among others were the main reason for irrational antibiotic prescription in India [3].

## Study limitation

It was not possible to link prescribers to the records retrieved. This level of care may have more challenge with antibiotic prescription. We did not assess diagnostic accuracy especially for upper respiratory tract infection; this might have resulted into over reporting of this condition and antibiotic prescription. We did not explore all the components of irrational prescription. Our sample size for the prescribers was not enough to establish the relationship between prescribers'/facilities factors and prescription practices.

## Conclusions

Antibiotics were mainly prescribed for the treatment of viral respiratory infections and acute watery diarrhea in children, conditions mainly caused by viruses making the antibiotic prescription irrational. We recommend regular refresher courses on regarding management of common childhood illness and, technical support supervision on adherence to antibiotic prescription guideline, and a qualitative study to explore factors influencing irrational antibiotic prescription practices.

## Supporting information

**S1 Dataset.**
(XLSX)

**S2 Dataset.**
(XLS)

## Acknowledgments

1) District Health Officer and Municipal Health Officer, Mbarara who permitted the study to be done in the District and Municipality respectively.

2) Prescribers who were the study respondents.

3) Department of Pediatrics and Child Health, Mbarara University of Science and Technology.

## Author Contributions

**Conceptualization:** Nelson Okello.

**Data curation:** Nelson Okello.

**Formal analysis:** Nelson Okello.

**Funding acquisition:** Celestino Obua.

**Investigation:** Nelson Okello.

**Methodology:** Nelson Okello.

**Project administration:** Celestino Obua.

**Resources:** Celestino Obua.

**Supervision:** Catherine Kyakwera, Elias Kumbakumba, Celestino Obua.

**Validation:** Joseph Oloro, Catherine Kyakwera, Elias Kumbakumba.

**Writing – original draft:** Nelson Okello.

**Writing – review & editing:** Nelson Okello, Joseph Oloro, Catherine Kyakwera, Elias Kumba-kumba, Celestino Obua.

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
