## [Decision Letter · Decision Letter 0]

28 Jul 2020

PONE-D-20-10851

Antibiotic Prescription Practices Among Prescribers for Children under fives at Public Health Centers IIIs and IVs in Mbarara District.

PLOS ONE

Dear Dr. Okello,

Thank you for submitting your manuscript to PLOS ONE. After careful consideration, we feel that it has merit but does not fully meet PLOS ONE’s publication criteria as it currently stands. Therefore, we invite you to submit a revised version of the manuscript that addresses the points raised during the review process.

We look forward to receiving your revised manuscript.

Kind regards,

Mehreen Arshad, M.D.

Academic Editor

PLOS ONE

Journal Requirements:

2. We note that a questionnaire was employed as a tool in this study. As such, we ask you to upload a copy of the questionnaire as a supplemental file.

4. Please amend your authorship list in your manuscript file to include author Joseph Oloro.

6. Please upload a copy of Figure 4, to which you refer in your text on page 15 and 25. If the figure is no longer to be included as part of the submission please remove all reference to it within the text.

Reviewers' comments:

Reviewer's Responses to Questions

**Comments to the Author**

1. Is the manuscript technically sound, and do the data support the conclusions?

Reviewer #1: Yes

Reviewer #2: Partly

Reviewer #3: Partly

2. Has the statistical analysis been performed appropriately and rigorously? 

Reviewer #1: Yes

Reviewer #2: Yes

Reviewer #3: No

3. Have the authors made all data underlying the findings in their manuscript fully available?

Reviewer #1: Yes

Reviewer #2: No

Reviewer #3: Yes

4. Is the manuscript presented in an intelligible fashion and written in standard English?

Reviewer #1: Yes

Reviewer #2: No

Reviewer #3: No

5. Review Comments to the Author

Reviewer #1: 1. Correct minor typographical errors (examples, page 12, line 12: 53.1 %5; on page 14, line 14: "contra indicated") and inconsistencies (e.g. capitalization of some diseases and drugs but not others)

2. The Uganda Clinical Guidelines 2016 and the WHO-IMNCI guidelines 2018 were referenced in the text, but not included in the references section

3. Statements are made in the text that URIs and acute watery diarrhea are mainly viral. Some readers may be aware that bacterial pathogens are occasionally implicated (e.g. Group A Strep or catarrhal Pertussis for URIs; ETEC, Shigella, etc. for diarrhea). To avoid confusion, it is recommended to cite the specific guidelines in the UCG 2016 (e.g. pages 777-778 for diarrhea in children), and what the recommendation is (e.g. erythromycin or ciprofloxacin for severe diarrhea or dysentery). This will help avoid confusion for readers who are not familiar with the prescribing guidelines, and give insight onto the findings of the study.

4. It may be worth mentioning in the text, when discussing the irrational prescribing of antibiotics for watery diarrhea, that a rotavirus vaccine campaign was undertaken in 2018.

5. In figure 2, it is recommended that the authors avoid using abbreviations for drugs and use full antibiotic names instead (e.g. Xpen, PPF are abbreviations that not all readers may recognize)

Reviewer #2: Thank you for the opportunity to review this paper. This study describes irrational antibiotic prescription within health centers in Uganda. The study design was a retrospective cross-sectional design and data was collected through a data extraction form and a questionnaire that was filled by investigators. As the study aims to “Our study aimed at determining antibiotic prescription practices of prescribers for children under five years at public health center IIIs and IVs in Mbarara District.” The adds to implementation of guidelines and describes antibiotic practices in Uganda. However, some concerns are raised in the reporting of the methodology of the paper, that requires major revisions. Additional information should be provided to fulfill the objective of this study.

1. In the abstract, page 7: “This may be associated with errors in antibiotic prescription. This practice has not been studied in our setting.”

This is better reformulated to clarify what “This” reflects.

2. Page 7: The term “health workers” is broad and is not accurate for describing antibiotic prescribing, since not all health workers prescribe them

3. Abstract, methods: This section needs to be reformulated, since it is not clear how the study was done. For example: “An interviewer administered questionnaire were used to collect health” looks like a third questionnaire different that the one introduced when interviewing health workers.

4. What is the difference between these “clinical officers and occasionally medical officers.”? The sentence starts with nurses being a profession then presumably compared to medical doctors?

5. Page 8: “to the lower health facilities within the district.” Why are these facilities considered lower?

6. The background introduction benefit from additional details on the availability of any studies antibiotic prescribing practices in Mbarara district or its health centers. If no previous studies exist, it should be highlighted as a gap in the literature.

7. In the entire methods section , it is better to separate the study into first 2 sections, one related to the reviewed records and with all its details, and the second related to the interviewed health providers and all its details. The authors switch back and forth with the two sections, which confuses the reader.

8. In the study limitations: “The data may not relate to the prescribers we interviewed”. The data in this case may be the data extracted from the records. However, the authors also interviewed the prescribers, and have data from the answers filled when they were interviewed them. I think this limitation is not of importance since they objective is not to link the two sources, rather each source should help in studying certain aspects of antibiotic prescribing.

9. It is mentioned in the study that health center 3 has a laboratory, shouldn’t it be capable of performing cultures? Please clarify

10. Since the authors did not cover all components of irrational drug prescribing (ex: cost), it cannot be concluded that prescribing was irrational, rather it should be described according to how irrational was measured: ex: inappropriate dose, indication, duration or period of use. These details on how the prescription was judged as irrational can be detailed if available in the data. The specific domains that were used to judge on how rational prescribing was should be mentioned. Also, since the objective of the study is to describe prescription practices, this should be present in the results; currently it is only reported that 68.4% of prescriptions are irrational.

Reviewer #3: There is significant quality research published in antibiotic resistance among children and its irrational use. Specific diseases have been highlighted and the overall antibiotic resistance has also been covered from low and middle income country settings. There is very little that the manuscript adds to existing knowledge base and is of little interest to wider readership. Examples of some recent literature include:

Tekleab AM, Asfaw YM, Weldetsadik AY, Amaru GM. Antibiotic prescribing practice in the management of cough or diarrhea among children attending hospitals in Addis Ababa: a cross-sectional study. Pediatric health, medicine and therapeutics. 2017;8:93.

Tasawer Baig M, Akbar Sial A, Huma A, Ahmed M, Shahid U, Syed N. Irrational antibiotic prescribing practice among children in critical care of tertiary hospitals. Pakistan journal of pharmaceutical sciences. 2017 Jul 2;30.

Hameed A, Naveed S, Qamar F, Alam T, Abbas SS, Sharif N. Irrational use of antibiotics. Different Age Groups of Karachi: a wakeup call for antibiotic resistance and future infections. J Bioequiv Availab. 2016;8:242-5.

Ilić K, Jakovljević E, Škodrić-Trifunović V. Social-economic factors and irrational antibiotic use as reasons for antibiotic resistance of bacteria causing common childhood infections in primary healthcare. European journal of pediatrics. 2012 May 1;171(5):767-77.

Sebsibie G, Gultie T. Retrospective assessment of irrational use of antibiotics to children attending in Mekelle general hospital. Sci J Clin Med. 2014 Jun 12;3(3):46-51.

Moreover, the manuscript is poorly written with weak references. Please note that the style of referencing for the journal is Vancouver style of referencing.

The article draws weak conclusions and is based on a study design which is lower in the hierarchy of evidence.

6. PLOS authors have the option to publish the peer review history of their article (what does this mean?). If published, this will include your full peer review and any attached files.

Reviewer #1: No

Reviewer #2: No

Reviewer #3: No

---

## [Author Response · Author response to Decision Letter 0]

20 Aug 2020

RESPONSE TO THE REVIEWERS

I would like to appreciate very much all your comments and guidance offered concerning my submission. I would therefore like to respond as follows;

REVIEWER 1

1. Typing error corrected as highlighted on page 12 line 12, page 14 line 14

2. Uganda Clinical Guideline 2016 (UCG 2016) and WHO – IMNCI 2012 included in the reference section as advised.

3. Statement about etiology of upper respiratory tract infections and acute watery diarrhea revised to indicate the possibility of bacterial etiology occasionally necessitating antibiotic therapy. 

4. Statement concerning Rota viral vaccine campaign to mitigate diarrhea has been added to the text.

5. Abbreviations have been removed from figure 2

REVIEWER 2

1. The statement on page 7 of the abstract has been reformulated.

2. I have revised the statement “health worker” so that it accurately describes the prescribers as highlighted.

3. The method section has been made clearer to indicate what we actually did as highlighted in the text.

4. In Uganda, Clinical officer is a title given to health cadres who completed a 3-year course leading to award of diploma in medicine and community health. They are the primary care prescribers in health centre IIIs, IVs, outpatient departments of District and Regional hospitals. Medical officers on the other hand are holders of Bachelor of Medicine and Bachelor of Surgery. In Uganda, Medical officers are employed at the level of health centre IVs and above. They work in both outpatient and inpatient departments.

5. Page 8. In Uganda, there is a policy on the level of care and referral systems. The lowest level being the Village Health Team (VHT), they provide home based care for malaria and pneumonia. VHTs identify and refer patients who may need further care to either Health centre II or III. Health centre II mainly handles vaccination and uncomplicated antenatal care. They refer their patient to health centre IIIs. Health centre IIIs refer to health centre IVs. Both health centre IIIs and IVs have basic laboratory services like microscopy and complete blood count only. Health centre IIIs and IVs can refer to a district hospital. The next level of care in the hierarchy are regional and national referral hospitals, they have capacity to offer specialized care. So, in Uganda from the level IV down to VHTs are referred to as lower level facility since they lack facilities for investigations.

6. There is no any study on antibiotic prescription practices in Mbarara and Uganda in general. This has already been highlighted as a gap that necessitated our study.

The method section has been separated to reflect records and prescribers’ interview as advised.

One limitation has been removed as suggested.

Health centre IIIs and IVs in Uganda have only basic laboratory services, they cannot perform microbiological studies including blood culture.

The conclusion that prescription of antibiotic was irrational in this setting has been revised to reflect components of irrationality assessed in our study.

Reviewer 3

I absolutely agree with you that a lot of literature exist on antibiotic resistance and irrational prescription. We however did not see any published article on antibiotic prescription practices in a rural setting of Uganda. We thought this study was necessary as a baseline for a bigger study with a better design which would inform policy on control and prevention of antibiotic resistance.

---

## [Decision Letter · Decision Letter 1]

1 Oct 2020

PONE-D-20-10851R1

Antibiotic Prescription Practices Among Prescribers for Children under fives at Public Health Centers IIIs and IVs in Mbarara District.

PLOS ONE

Dear Dr. Okello,

Thank you for submitting your manuscript to PLOS ONE. After careful consideration, we feel that it has merit but does not fully meet PLOS ONE’s publication criteria as it currently stands. Therefore, we invite you to submit a revised version of the manuscript that addresses the points raised during the review process.

Please read the comments below carefully. There appear to be many grammatical errors which need to be corrected. The references also appear to be in the wrong format. 

We look forward to receiving your revised manuscript.

Kind regards,

Mehreen Arshad, M.D.

Academic Editor

PLOS ONE

Additional Editor Comments (if provided):

Thank you for responding to the reviewer comments. In reviewing further there are several other clarifications that need to be as well. Please also read through the entire manuscript carefully. There are many grammatical errors that need to be corrected.

1. Introduction para 2: please explain the different clinical officer and medical officer

2. Intro para 4: please explain the difference between HC IIIs and IVs

3. Methids: Sample size and sampling procedure: Please read through this paragraph carefully, there are many grammatical errors in it

4. It is unclear why there are 2 paragraphs under the heading 'Results from prescribers’ interview' under the tables and figures section.

5. The legends for the tables and the figures need to be made more concise. They should not describe what is already noted in the table/figure. Also the legends appear to be referencing the figures incorrectly. For eg. the legend above figure 2 references figure 3.

Reviewers' comments:

Reviewer's Responses to Questions

**Comments to the Author**

1. If the authors have adequately addressed your comments raised in a previous round of review and you feel that this manuscript is now acceptable for publication, you may indicate that here to bypass the “Comments to the Author” section, enter your conflict of interest statement in the “Confidential to Editor” section, and submit your "Accept" recommendation.

Reviewer #1: All comments have been addressed

2. Is the manuscript technically sound, and do the data support the conclusions?

Reviewer #1: Yes

3. Has the statistical analysis been performed appropriately and rigorously? 

Reviewer #1: Yes

4. Have the authors made all data underlying the findings in their manuscript fully available?

Reviewer #1: Yes

5. Is the manuscript presented in an intelligible fashion and written in standard English?

Reviewer #1: Yes

6. Review Comments to the Author

Reviewer #1: (No Response)

7. PLOS authors have the option to publish the peer review history of their article (what does this mean?). If published, this will include your full peer review and any attached files.

Reviewer #1: No

---

## [Author Response · Author response to Decision Letter 1]

5 Nov 2020

1. Introduction para 2: In Uganda, Clinical officer is a title given to health cadres who completed a 3-year course leading to award of diploma in medicine and community health. They are the primary care prescribers in health centre IIIs, IVs, outpatient departments of District and Regional hospitals. They literally work as assistant medical officers. Medical officers on the other hand are holders of Bachelor of Medicine and Bachelor of Surgery. In Uganda, Medical officers are employed at the level of health centre IVs and above. They work in both outpatient and inpatient departments.

2. Introduction para 4: In Uganda, there is a policy on the level of care and referral systems. The lowest level being the Village Health Team (VHT), they provide home based care for malaria and pneumonia. VHTs identify and refer patients who may need further care to either Health centre II (level 2) or III (level 3). Health level 2 mainly handles vaccination and uncomplicated antenatal care. They refer their patients to health centre IIIs (level 3). Health centre IIIs refer to health centre IVs (level 4). Both health centre IIIs (level 3) and IVs (level 4) have basic laboratory services like microscopy and complete blood count only. Health centre IIIs and IVs can refer to a district hospital. The next level of care in the hierarchy are regional and national referral hospitals, they have capacity to offer specialized care. So, in Uganda from the level IV down to VHTs are referred to as lower level facility since they lack facilities for investigations.

3. Methods: I have addressed the grammatical errors noted under method section, refer to page 4 and 5 in the text.

4. The double heading ‘Results’ and legends for tables and figures have been revised to avoid repetition. 

5. The citations style has been changed to ‘plos’ format

---

## [Editor Report · Decision Letter 2]

30 Nov 2020

Antibiotic Prescription Practices Among Prescribers for Children under fives at Public Health Centers IIIs and IVs in Mbarara District.

PONE-D-20-10851R2

Dear Dr. Okello,

We’re pleased to inform you that your manuscript has been judged scientifically suitable for publication and will be formally accepted for publication once it meets all outstanding technical requirements.

Kind regards,

Mehreen Arshad, M.D.

Academic Editor

PLOS ONE

---

## [Editor Report · Acceptance letter]

11 Dec 2020

PONE-D-20-10851R2 

Antibiotic Prescription Practices Among Prescribers for Children under five at Public Health Centers III and IV in Mbarara District. 

Dear Dr. Okello:

I'm pleased to inform you that your manuscript has been deemed suitable for publication in PLOS ONE. Congratulations! Your manuscript is now with our production department. 

Kind regards, 

on behalf of

Dr. Mehreen Arshad 

Academic Editor

PLOS ONE